# Digital Marketing Enhancement of Cryptocurrency Websites through Customer Innovative Data Process

**Damianos P. Sakas** [1], **Nikolaos T. Giannakopoulos** [1,*], **Nikos Kanellos** [1] **and Christos Tryfonopoulos** [2]

1   Business Information and Communication Technologies in Value Chains Laboratory (BICTEVAC LABORATORY), Department of Agribusiness and Supply Chain Management, School of Applied Economics and Social Sciences, Agricultural University of Athens, 118 55 Athens, Greece; d.sakas@aua.gr (D.P.S.); nikos.kanellos2@aua.gr (N.K.)
2   Department of Informatics & Telecommunications, University of Peloponnese, Karaiskaki 70, 221 00 Tripoli, Greece; trifon@uop.gr
*   Correspondence: n.giannakopoulos@aua.gr; Tel.: +30-694-001-3673

**Abstract:** Today, more than ever, the popularity of decentralized payment systems has risen, creating an outbreak of new cryptocurrencies hitting the market. Unique websites have been staged for each cryptocurrency, where information and means for mining cryptocurrencies are available daily. People visit those cryptocurrency websites either from desktop or mobile devices. Thus, the impulsion for appropriate promotion of cryptocurrency websites and customer factors affecting it rises. The above process increases cryptocurrency organizations' website visibility, raising the need for customer relationships and satisfaction optimization concerning organizations' supply chain strategy. Research data were collected from 10 well-known cryptocurrency websites, regarding mobile and desktop devices, in 180 days, regarding on-site web analytics. Therefore, a model consisting of three stages was applied. Starting phase of the model pertains to statistical and regression analysis of cryptocurrency web analytics, followed by Fuzzy Cognitive Mapping and Agent-Based Model deployment. Throughout this study, methods for promoting cryptocurrency websites can be deduced from assessing specific website metrics and device preferences. Research results indicate that web analytics give a clearer image of customer behavior in cryptocurrency websites and, therefore, provide opportunities for further website optimization through increased web traffic and digital reputation.

**Keywords:** strategic digital marketing; innovation process; decentralized systems; data analysis; web analytics; customer electronics; decision support systems

## 1. Introduction

Nowadays, the hype about cryptocurrencies has led to the development of more than 1600 cryptocurrencies. Cryptocurrencies are digital coins that may be transferred online. Cryptographic encryption and digital certificates are used to validate transactions and prevent multiple spending of the very same coin. By forbidding users from replicating the data that form the coin, cryptocurrencies have brought the concept of shortage to the digital realm [1]. Cryptocurrencies might become lucrative because their shortage is maintained by the encryption built in their transparent code (normally auditable by anybody). In opposition to fiat currency, bitcoin is produced, exchanged, circulated, and preserved via a decentralized registration process and is referred to as a blockchain. The Bitcoin value is influenced by a wide range of factors, including global opinion, media, and buzz [2].

Despite Bitcoin being still the most popular cryptocurrency, many other digital currencies have already been created. Cryptocurrency commodities are divided into cryptocurrencies such as Ethereum, Ripple, and Dogecoin; stable coins such as Binance USD and Tether; and tokens [3]. Bitcoin contains characteristics that were not held by traditional financial transaction channels, such as the fact that the Bitcoin price fluctuates based on

people's perceptions and views as well as institutional practices. The increasing volatility of the crypto value leads to risky currency transactions [4].

Nonetheless, this ever-expanding financial sector is marked by substantial volatility and sharp price variations over time. Today, cryptocurrency prediction is widely regarded as one of the most difficult time-series forecastings, given the multitude of unpredictability variables involved as well as the considerable volatility of cryptocurrency prices, culminating in complex chronological correlations [5,6]. With the exception of fiat currencies, where the government may alter supply to counteract price bubbles, the distribution of a cryptocurrency is often built to meet a predefined route, making it more susceptible to price variations. The additional network factor of a cryptocurrency is a second crucial property. The advantage of acceptance for a user is determined by the quantity of all other users who can always trade [7].

While blockchain technologies were first used in the framework of bitcoin, they have now spread to regular enterprises, entrepreneurs, and everyday activities. It will be used, for instance, to provide an alternate payment solution to credit cards or PayPal in e-commerce and worldwide transactions [8]. Furthermore, it has been incorporated in financial institutions, as well as many other industries, with diverse applications from facilitating and standardizing financial intermediation to monitoring trader loyalty cards, and even constructing decentralized markets for commercial transactions [9]. It is already being used by Volkswagen and others for electricity exchange and grid management, as well as by mobile firms like Huawei and Apple that are creating blockchain-enabled devices to allow customers to pay in bitcoin through smartphones [10].

Big Data and web analytics are key factors of our research, enabling analysis of elements that affect the website's performance, like user engagement metrics. Big data is defined as a vast amount of information with both a massive density and heterogeneity, whose velocity surpasses existing technology's ability to manage correctly [11]. Web analytics entails collecting and analyzing massive volumes of data designed to check and enhance organizations' site web usage [12]. Metrics are employed by web analytics systems to reduce web traffic data to plain values which are easy to understand. Customers utilize trade credit as a financial management tool to keep their organization's current liquidity in check. From a company's perspective, trade credit enables them to create an attractive payment schedule without jeopardizing their profitability [13].

The blockchain method is suitable for settings that demand high levels of verification and validation, and it can also respond to environmental modifications and regulatory measures, such as governing agencies [14]. Cryptocurrencies are a relatively new payment option that provides a competitive edge to company websites [15]. As a result, Key Performance Indicators (KPI) based on web analytic metrics [12] are a valuable measurement for evaluating site goal completion. Numerous indicators and KPIs are used in web analytics to collect internet usage information in intelligible and convenient ways. Advertisers and analysts may use several methods to increase customer engagement [16], brand recognition [16,17], and profitability [16,17], in addition to the digital payment alternatives that websites provide to users, increasing their value [18].

The main interest of this research was how the exploration of cryptocurrency website customers' engagement metrics affects their organic traffic and global rank. Cryptocurrency organizations should pay more attention to the behavior of their websites' customers; hence there is great potential for further digital promotion. Since they could present the effectiveness of their website's digital promotion campaigns through increased organic traffic and enhanced global rank, customers' web analytics are key metrics for evaluation.

### 1.1. Digital Marketing of Cryptocurrency Websites

#### 1.1.1. Importance of Customer Engagement and Device Preference in E-Commerce

Besides obtaining more attention, digital marketing advertises products and services through mobile applications and business websites [19]. As a consequence, corporate digital brand name and exposure will improve. Marketers use technological tools to

provide a variety of direct and online advertising to attract customers' engagement and enhance client loyalty [20,21]. Web analytics in digital advertising, for instance, enables you to customize the customer experience [22]. The overall value of the service supplied on corporate websites, as well as their compatibility with the targeted customers, influences website traffic, customer engagement, and visibility [23].

Within this framework, the parameters under which cryptocurrency traders and customers mutually reinforce one other in terms of sustaining risky profits and retention advantages, respectively. On the one side, customers profit from the network spillovers produced by speculative investors, promoting user engagement. Traders anticipating a steadier flow of users to acquire cryptocurrency, on the other side, might not even have to depend as largely on the faith of their other traders to secure a return. Wei & Dukes [6] suggest that if an event raises the core of prospective users, cryptocurrency inflation is more likely to arise (yet less likely to fall when created).

Cryptocurrencies highlight the necessity for greater practical research and exploration into virtual currency, with recent studies concentrating on individual intentions to embrace cryptocurrency [24] for various reasons such as payment, etc. Furthermore, the majority of the existing literature globally ignores end-user engagement parameters, i.e., individual customer level [25], as the innovation of cryptocurrencies could be dependent on explaining customer behavior and also predicting the drivers that can enhance the engagement process in each device (mobile and desktop).

### 1.1.2. Digital Promotion and Cryptocurrency Data Analysis

Digital promotion, big data, and web analytics together serve an essential function in the development and sustainability of a company's digital brand name, as well as profitability [16,26]. Our study concentrated on statistics relating to bitcoin webpage visibility, as measured by analytic metrics across their domains. Furthermore, one major issue that should be investigated properly in the coming years is the use of cryptocurrency relevant data such as estimated average price, opening and daily closing price, maximum and minimum everyday rates, the everyday amplitude of transactions, and perhaps even financial and technical trading measures [27].

Social networks may be utilized to discover what people are thinking about commodities, events, needs, and supplies. Twitter, a well-known social media platform, enables its users to express themselves and contribute information that influences the market situation [28]. As a result, expression analysis is critical for recognizing and comprehending adaptive and maladaptive user demands [29,30]. Cryptocurrency price fluctuations are strongly influenced by social networking attitudes, and analysis is based on online search techniques. Although individuals often Tweet favorably about cryptocurrencies as their values fall, Twitter attitude research believes forthcoming price levels will become favorable. As a result of their volatile nature in the current market, predicting cryptocurrency values is a difficult endeavor [31].

We investigated the drivers of behavioral intention [32], among the most dominating dependent variable in engagement-related research [33]. Per the results of their study, performance expectation and price value have a favorable impact on cryptocurrency website customers' behavioral intention to use bitcoin. This suggests that customers are more likely to accept cryptocurrencies if they perceive that using this innovation is beneficial and allows individuals to execute a task effectively. Furthermore, it is significantly confirmed the link involving performance expectancy with a market value in the context of bitcoin engagement. People see cryptocurrencies as a good technology that has a substantial influence on their daily lives and provides massive advantages (for example, simplicity, time savings, and effectiveness). They also consider engaging in cryptocurrency is more profitable than acquiring this technology [34].

### 1.2. Decentralized Systems of Payment and Customer Website Visits

1.2.1. Decentralization as an Innovative Strategical Payment Tool

Even the notion of innovation is not new. It has been estimated that it is as old as humans [35]. It is self-evident that all sorts of innovation do have a positive influence on the overall sustainability of enterprises and, by extension, economies. Implementing a new or significantly improved product (goods or services) or practice, a new advertising technique, or a new organizational strategy in company operations, workplace organization, or external relations is characterized as an innovation [36]. Companies employ innovation to gain a competitive edge, and it is a vital ingredient of commerce [37]. Blockchain technology includes many innovative elements in both its structure and operation.

A blockchain is an innovation that has moved to the forefront for having a safe, protected, and secret online identity. Distributed ledger technology (DLT), the most well-known example being blockchain, make sure that the information has never been maintained in a centralized file and instead is privately handled in decentralized networks. This would give individuals control of their identification by producing a universal ID that can be used for many reasons [38]. Furthermore, once the payments are stored in the blockchain, they cannot be altered. As more than just a result, it also can change many industries, including financial products [39], online payments [40], government services [41], Internet of Things (IoT) [42], brand image structures [43], and security agencies [44]. All participants on the blockchain network can track down the payments. So, each transaction stored on the blockchain may be verified as genuine. Nonetheless, the person involved in the trade is undetermined [45].

The potential of smart contracts to expedite paperwork and transaction procedures boosts productivity and consequently decreases the costs, which is thought to be simple to apply [46]. This is consistent with Kamble et al. [46] findings that observed engagement seems to be the primary driver underlying blockchain technology acceptance. Human ingenuity as a monitor has been effective. The moderate outcome implies that inventive customers would try to utilize and adapt cryptocurrencies even if they do not believe their worth is great (e.g., price value) [34].

Mobile transactions, which may not necessitate a desktop to make the transaction and are occasionally invoiced through the telecommunication provider, are another payment option [47]. Because of the launch of mobile trading software and its unique marketing approach, the aforementioned payment mechanism has seen exponential firm growth since its creation [48]. Blockchain enables payments to be made without the participation of a 3rd party, including a bank or PayPal, but it can be used for a broader range of financial products [49].

1.2.2. KPIs for Cryptocurrency User Engagement and Website Traffic Sources

Keeping their demands and expectations of currency purchasing in view, users' incentive on evaluated performance expectancy and perceived price of cryptocurrencies offer an important study [50]. Consequently, the features of cryptocurrency websites need to include: faster page production, higher user satisfaction, continual updates, mobile applications or mobile-friendly websites, etc. The webpage system's emphasis is on the frequency of updates at the rate of data collection from Web trading platforms and social networks. [51]. Therefore, the use of blockchain for better marketing and advertising has simultaneously boosted customer privacy [52].

Cryptocurrencies, for example, might offer long-term potential, especially if they encourage a speedier, more reliable, and more effective payment system [53]. However, while previous studies considered cryptocurrencies primarily as cash instead of technology, researchers hardly evaluated the link between innovation capability and bitcoin pricing. When considering cryptocurrency as an innovation, it presupposes that it will have a variety of use cases and implementations (e.g., payments, smart contracts, data gathering) that can generate a certain level of value [54].

Most customers arrive at websites via different channels such as direct, referral, search, paid, and social. The breakdown of website analytic customer engagement data, such as bounce rate, average time on site, and average pages per visit, may be used to assess digital marketing and promotion tactics [55]. To measure their digital promotion efficacy, cryptocurrency firms should study their monthly-tracked website performance indicators, which include worldwide rank and organic traffic, as well as analytic data such as average pages per visit and bounce rate. Key Performance Indicators (KPIs) are a quantitative indicator of performance over time for a given goal and must meet specified standards for web analytics and digital marketing [41]; consequently, authors describe, depict, and assess the impact of the chosen KPI rates each month. Table 1 depicts the investigated KPIs for this study.

**Table 1.** Description of the examined web metrics.

| Web Analytics/KPIs | Description of the WA/KPIs |
|---|---|
| Global (Web) Rank/month | The relatively less ranking a website has the more visibility it receives, since it depicts the ratings of all websites across the internet in terms of customer popularity. Web Rank measures a website's attractiveness to other websites, providing it with a helpful KPI for comparative analysis [56]. |
| Organic Traffic/month | One of the most powerful markers of SEO effectiveness is the tracking of organic sessions over time. A month-over-month rise in organic search visits indicates that a website's rankings are increasing. While other metrics may reveal trends, this statistic provides concrete evidence that the organization's efforts bring in more visits, prospects, and, hopefully, conversions [57]. |
| Traffic Sources/month | Because the purpose of a website is the creation of traffic and leads, knowing how people arrive at a site is the most important tool in improving it. So, traffic sources refer to the way customers reach your website and expanded at direct, referral, search, paid, and social traffic sources. [58]. |
| Returning Visitors/month | Visitors who have recently visited a website are referred to as returning visitors. When a visitor returns to a webpage after two years, they are considered a returning visitor [59] |
| Unique Visitors/month | A unique visitor is defined as someone who views a webpage once in a referred period. Unique visitor numbers reveal how many people a website truly attracts [59]. |
| Bounce Rate/month | The percentage of website visitors who leave after seeing a single page is called the bounce rate. The bounce rate shows how good the content is, so it has to be kept at a low rate [59]. |
| Average Time on Site/month | The time spent on site is calculated by dividing the duration of an average visit during a certain time period by the total number of visits over that period [59]. |
| Average Pages per Visit/month | Pageviews have been the most fundamental of all user engagement metrics, tracking the number of times a user visits a specific page on a website [59]. |

*1.3. Research Motivations*

The specter of digital marketing's potential strategies and methodologies for improving an organization's financial performance is vast. Most importantly nowadays, due to the volatility of the financial markets and especially cryptocurrency markets, where organizations promote their currencies and projects via their websites, the need for daily and intraday data is vital. This aspect can be satisfied by utilizing website analytics, which belongs to the Big Data family and provides notable insights for intraday and daily analysis of website user behavior. Authors focused on analyzing intraday website behavioral

analytics of the top cryptocurrency organizations and attempted to assess the efficiency of their digital marketing efficiency with the engagement of their website customers.

Outcomes of this research could administer valuable information to cryptocurrency organizations regarding the importance of enhanced digital marketing strategies in their profitability. Therefore, solutions could be given to key questions, such as how intraday and daily website customer data could be retrieved, whether those data are connected with organizations' digital marketing campaigns, and which data could enhance cryptocurrency organizations' digital marketing strategies.

### 1.4. Related Literature

Recent studies of related subjects indicated the plethora of applications blockchain technology has in many research areas. Rejeb et al. [60] stated there is a general interest in examining new technologies like blockchain, implications in evaluating marketing, and corporate brand names. Moreover, the investigation of cryptocurrency users can produce knowledge patterns for explaining the phenomenon of speculative bubbles [61]. Classification of cryptocurrency websites' customers can be performed by applying past modeled user behavior [62], adding more knowledge to the existing blockchain literature. Visual analytics significantly contributed to analyzing the marketing communication and reputation of cryptocurrencies, highlighting the role of social analytics [63].

Generally, more research should be focused on blockchain derivatives and digital marketing strategies evaluation. A research gap can be found regarding the connection of digital marketing in cryptocurrencies, as blockchains' implications on finance, and the effect of website customers' analytic data. There should be shed more light on whether the utilization of customers' website behavior could benefit cryptocurrency organizations and thus, boost their digital marketing efficiency.

### 1.5. Approach of the Study

By elaborating on the literature review, combined with the requirement of examining cryptocurrencies' customer behavior, the authors propose the following methodological approach. In Figure 1, the complete mapping of the analyzed process of the approach is shown. The thinking behind the paper's approach is based on potential cryptocurrency website visitors and the path they follow to end up on those websites, combined with the way their behavior affects cryptocurrency organizations' digital marketing processes. A potential customer should enter a cryptocurrency website through a desktop or a mobile device. From each device, a specific path could be chosen to enter cryptocurrency websites, reflecting 5 possible ways (direct, referral, search, social, and paid traffic). Once landing on the website, customers interact with it, leaving their unique behavioral pattern. Cryptocurrency organizations could utilize these patterns (device preference, source of traffic, and website behavioral analytics) to enhance their digital marketing efficiency and thus increase their profitability at a later phase.

### 1.6. Research Hypotheses

Over recent years, the cryptocurrency market has known a vigorous increase in reputation and has expanded vastly. Increased reputation and market capital explain the growing number of customers entering cryptocurrency websites has risen accordingly. The means of entering cryptocurrency websites and the analytic metrics of each user's web visit account for unique customer behavior motifs. Traffic sources that customers choose to enter a website through consist of 5 different types, all of which have different attributes, making them more disguisable. Anent to customer engagement, it also consists of various user web metrics that combined characterize the behavior of a website customer. For cryptocurrency organizations to acknowledge this detailed information over their website customers and estimate their impact on their website visibility and digital brand name could possess significant advantages. In particular, the paper's contribution to the field intends to subserve:

- Cryptocurrency marketers and strategists define organizations' goals and advancement in the market through evaluating website customer behavior.
- Developers of cryptocurrency websites adjust and upgrade aspects of their cryptocurrency platform operations to attract and engage more customers.

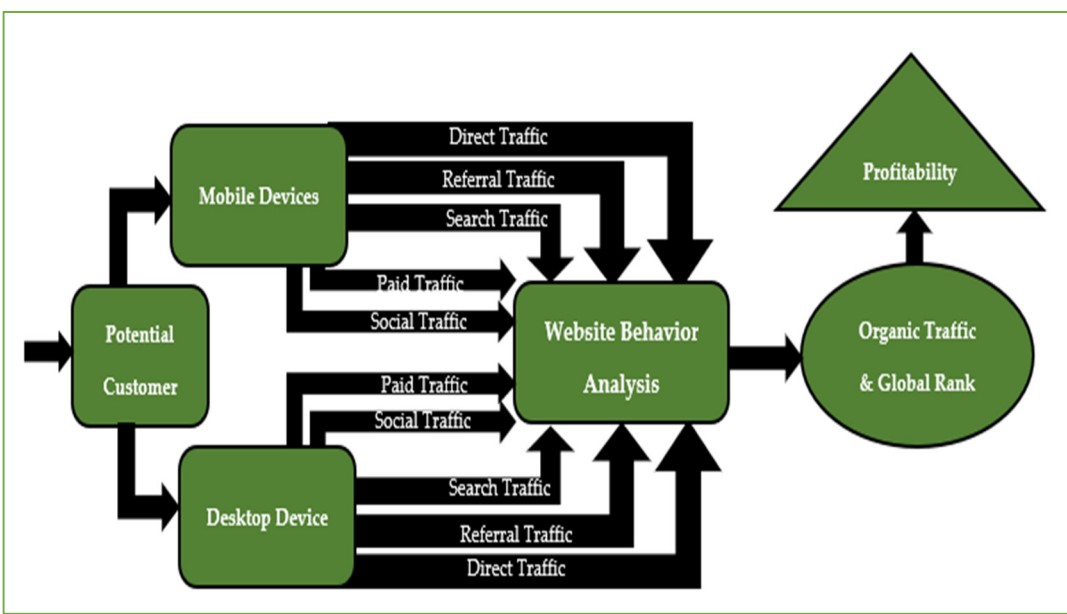

**Figure 1.** Conceptual Framework of Cryptocurrency websites.

In accordance with the above, the authors settle the ensuing research hypotheses to investigate the effect of customer engagement metrics and traffic sources on cryptocurrency organizations' digital marketing efficiency.

**Hypothesis 1 (H1) .** *Cryptocurrency website Global Rank gets significantly affected by Mobile and Desktop Device Customer Engagement metrics.*

The placement of the research's first hypothesis aims to discover the significance of mobile and desktop customer engagement metrics' impact on cryptocurrency websites' global rank. In this way, authors could determine whether analyzing the engagement metrics of website customers, from either device (desktop or mobile), to estimate cryptocurrency website global rank.

**Hypothesis 2 (H2).** *The impact of Mobile and Desktop Device Customer Engagement metrics on Cryptocurrency website Organic is significant.*

Moving to the second hypothesis, the authors aim to measure again the effect that causes desktop and mobile device customer engagement metrics on cryptocurrency websites organic traffic to understand the capability of engaged customers and device preferences that can lead to more organic traffic to cryptocurrency websites.

**Hypothesis 3 (H3).** *Web Traffic Sources pose a significant effect on Cryptocurrency websites. Global Rank gets significant.*

In the third hypothesis, the impact of potential cryptocurrency website customer traffic sources (direct, referral, paid, search, and social traffic) on their global rank is investigated. Examining the significance of the combined traffic sources effect on cryptocurrency websites' global rank could provide us with a handful of information on their relationship.

**Hypothesis 4 (H4).** *The impact caused on Cryptocurrency website Organic Traffic by Customer Web Traffic Sources is significant.*

In the same context, calculation of the impact caused to cryptocurrency websites' organic traffic from the combination of the 5 traffic sources could, if flagged important, indicate that specific marketing strategies regarding traffic sources draw in to cryptocurrency websites more organic customers.

**Hypothesis 5 (H5).** *Individual analysis of web Traffic Sources has significant correlations with Cryptocurrency website Global Rank or Organic Traffic.*

Regarding our paper's last hypothesis, our objective is to strengthen Hypothesis 4 by setting the parameter of individual traffic sources' significant impact on either cryptocurrency website global rank or organic traffic. From this hypothesis, the important traffic sources for cryptocurrency websites' global rank or organic traffic will be discerned, rendering organizations precious knowledge over digital promotion potentialities.

## 2. Materials and Methods

One of the main aims of this paper is to propose a methodological framework for organizations falling into the cryptocurrency market regarding efficient digital marketing and promotion techniques. Initially, gathering the necessary big data from cryptocurrency websites through web analytic tools, and afterward analyzing them, are correct steps concerning the main goal. In this way, the intercorrelations and relationships between the sample variables can be disguised. To acknowledge cryptocurrency digital promotion variables, the authors identified the importance of organic traffic, global rank, and user engagement metrics, such as bounce rate, average time on site, average pages per visit, returning, and unique customers.

Our framework analyzes and develops an exploratory and diagnostic model for estimating the above relationships of the paper's examined metrics. The capitalization of macro-level analysis provided by Fuzzy Cognitive Mapping (FCM) [64] is essential. Through FCM, the impact of each web metric on the chosen variables can be depicted and then, via regression analysis, calculated. Thus, in this framework stage, our analysis aims to give cryptocurrency organizations valuable data for possible implications of specific web metrics to their digital brand name. The alignment of the paper's aims with the appropriate KPIs extracted from data collection is implied.

The methodology's next phase emphasizes harvesting the statistical data eventuated from the previous stage to predict and estimate the course of data in the following time. Simulation model development will be accomplished by implementing a micro-level analysis, like Agent-Based Modelling [65], and the model inputs are comprised of regression coefficients and metrics' correlations. A key result of implementing the referred framework is the ability provided to cryptocurrency organizations to develop a more efficient digital marketing strategy for their websites based on their web metrics.

*Sample Selection and Data Availability*

For the next phase of the paper, data were gathered from a chosen sample of cryptocurrency organizations' websites based on the appeal their currency has to potential and present traders [66]. The main reason for choosing to analyze web analytics is their capability of calculating vast amounts of data, the accuracy of research results, and the fact that, for the research, a sampling of 3,000,000 website behavioral data. In so doing, the authors collected web data and analytic metrics on on-site customer behavior, widely known as Web Analytics, consisting of metrics such as the rate of abandoning a website, the potentiality of returning to a website again and again, etc. Those data were collected daily from various Decision Support Systems (DSS), provided by specific website platforms, for 180 consecutive days, reflecting 6 months. Sample web analytics refers to the undermen-

tioned cryptocurrencies' websites: Bitcoin, Ethereum, Binance, Tether, Cardano, Dogecoin, Ripple, Polkadot, Internetcomputer, and Bitcoincash.

By data gathering, authors report daily analysis and examine the chosen web metrics, aiming to comprehend customer behavior better. In a 180-days observation period, data were monitored and collected, with the upcoming estimation and prediction period ranging up to 360 days. A suggestion to cryptocurrency organizations could be the monitoring and assessment of their global rank and organic traffic as key factors for evaluating digital marketing strategies' efficiency. This could assist in costs reflecting big data collection and resources management regarding potential users' digital promotion strategies.

## 3. Results

As we move to the 3rd section of the paper, we see in Table 2 the results of the descriptive statistics performed on the 10 Cryptocurrency websites used in data collection. The statistics of mean, max, min, and std deviation are discerned.

**Table 2.** Descriptive Statistics of the 10 airline companies' websites during a six-month period.

|  | Mean | Min | Max | Std. Deviation |
|---|---|---|---|---|
| Global Rank | 36,553.34 | 25,566 | 54,613 | 12,367.9 |
| Organic Traffic | 9,754,603.8 | 8,879,592 | 10,951,569 | 705,527.3 |
| Direct Traffic | 33,925,162 | 110,000,000 | 71,886,776.83 | 28,314,776.8 |
| Referral Traffic | 3,519,269 | 14,417,648 | 8,760,133.5 | 4,336,519.4 |
| Paid Traffic | 8142 | 24,224.00 | 16,619.16 | 6354.67 |
| Social Traffic | 727,118 | 2,034,309 | 1,466,274.3 | 4,81,121.66 |
| Search Traffic | 9,133,353 | 29,705,871 | 20,526,018.66 | 7,367,251.26 |
| Bounce Rate Desktop | 0.64 | 0.70 | 0.6852 | 0.0256 |
| Averages Pages/Visit Desktop | 2.01 | 2.15 | 2.09 | 0.0507 |
| Average Time on Site Desktop | 439.9 | 502.6 | 478.7 | 26 |
| Unique Visitors Desktop | 7,205,729 | 30,301,774 | 18,821,029.5 | 8,444,914.5 |
| Return Visitors Desktop | 37,680,624 | 140,029,423 | 90,034,358 | 37,861,191.9 |
| Bounce Rate Mobile | 0.57 | 0.67 | 0.6475 | 0.0391 |
| Averages Pages/Visit Mobile | 2.05 | 2.61 | 2.4841 | 0.21744 |
| Average Time on Site Mobile | 355.90 | 417.50 | 392.4500 | 22.18547 |
| Unique Visitors Mobile | 3,194,633 | 5,012,216 | 4,009,864 | 665,940 |
| Return Visitors Mobile | 9,644,389 | 15,130,931 | 12,451,188.1 | 2,184,037.4 |

N = 180 observation days for 10 cryptocurrency websites.

Next, in Tables 3 and 4, we get the results of the Cronbach Alpha and KMO tests performed to measure tolerance levels and item coherence. Levels near 0.7 and higher indicate that variables are ideal for statistical analysis [67,68], so our combined variables for mobile and desktop engagement metrics are proper for regression analysis.

**Table 3.** Internal Consistency of Desktop Engagement.

|  | Cronbach's Alpha | Kaiser-Meyer-Olkin Factor Adequacy | % of Total Variance Explained |
|---|---|---|---|
| Desktop Engagement | 0.689 | 0.661 | 78.607 |

**Table 4.** Internal Consistency of Mobile Engagement.

|  | Cronbach's Alpha | Kaiser-Meyer-Olkin Factor Adequacy | % of Total Variance Explained |
|---|---|---|---|
| Mobile Engagement | 0.706 | 0.718 | 83.139 |

Tables 5 and 6 refer to the impact of mobile and desktop engagement metrics on cryptocurrency websites' global rank. Through linear regression modeling, we see that all variables affect statistically significant the global rank dependable variable with *p*-levels of significance below 0.01. Mobile and desktop engagement metrics' significant regression had $p = 0.000$ and $R^2 = 1.00$ in each case, respectively, with both regressions being verified. Cryptocurrency websites' global rank potential variation rates up to −0.277 from mobile bounce rate, −0.656 from average pages per visitor, 0.648 from average time on site, 4.459, and −4.352 from unique and returning visitors, respectively. Desktop engagement metrics variate global rank up to −2.160 (bounce rate), −1.891 (average pages per visit), 3.106 (average time on site), −3.579 (unique visitors), and 1.530 (returning visitors).

**Table 5.** Impact of Mobile Engagement metrics on Cryptocurrency websites' Global Rank.

| Variables | Standardized Coefficient | $R^2$ | F | *p* Value |
|---|---|---|---|---|
| Constant | 0 | | | 0.000 ** |
| Bounce Rate | −0.277 | | | 0.000 ** |
| Average Pages/Visit | −0.656 | | | 0.000 ** |
| Average Time on Site | 0.648 | 1.00 | - | 0.000 ** |
| Unique Visitors | 4.459 | | | 0.000 ** |
| Returning Visitors | −4.352 | | | 0.000 ** |

** Indicate statistical significance at the 99% level.

**Table 6.** Impact of Desktop Engagement metrics on Cryptocurrency websites' Global Rank.

| Variables | Standardized Coefficient | $R^2$ | F | *p* Value |
|---|---|---|---|---|
| Constant | 0 | | | 0.000 ** |
| Bounce Rate | −2.160 | | | 0.000 ** |
| Average Pages/Visit | −1.891 | | | 0.000 ** |
| Average Time on Site | 3.106 | 1.00 | - | 0.000 ** |
| Unique Visitors | −3.579 | | | 0.000 ** |
| Returning Visitors | 1.53 | | | 0.000 ** |

** Indicate statistical significance at the 99% level.

For every 1% of increase in mobile bounce rate, average pages per visit, average time on site, unique and returning visitors, cryptocurrency websites' global rank decreases by 21.7%, decreases by 65.6%, increases by 64.8%, increases by 445.9% and decreases by 435.2% accordingly. As desktop bounce rate, average pages per visit, average time on site, unique and returning visitors increase by 1%, cryptocurrency websites' global rank decreases by 216%, decreases by 189.1%, increases by 310.6%, decreases by 357.9% and increases respectively by 153%. The outcomes of the first two regressions verify our first research hypothesis that cryptocurrency websites' global rank is significantly affected by mobile and desktop engagement metrics.

Tables 7 and 8 show the regression of cryptocurrency websites' organic traffic with desktop and mobile engagement metrics. Both regression models appear to be statistically significant, with *p*-levels = 0.000 and $R^2 = 1.00$. More specifically, organic traffic of cryptocurrency websites variates up to −0.541 from desktop bounce rate, −0.763 from average pages per visitor, 1.776 from average time on site, 2.504 and −3.160 from unique and returning visitors, respectively. As for mobile engagement metrics, they cause a variation in global rank up to −1.037 (bounce rate), −1.095 (average pages per visit), 0.217 (average time on site), −2.555 (unique visitors), and −1.665 (returning visitors). This means that when the desktop bounce rate, average pages per visit, average time on site, and unique and returning visitors increase by 1%, cryptocurrency websites' organic traffic decreases by 54.1%, decreases by 76.3%, increases by 177.6%, increases by 250.4% and decreases by 316% accordingly, while by every 1% rise of mobile bounce rate, average pages per visit, average time on site, unique and returning visitors, organic traffic increases by 103.7%, decreases by 109.5%, increases by 21.7%, increases by 255.5% and decreases correspond-

ingly by 166.5%. These results verify our second research Hypothesis, which assumes that Cryptocurrency websites' organic traffic gets significantly affected by desktop and mobile engagement metrics.

**Table 7.** Impact of Desktop Engagement metrics on Cryptocurrency websites' Organic Traffic.

| Variables | Standardized Coefficient | $R^2$ | F | *p* Value |
|---|---|---|---|---|
| Constant | 0 | | | 0.000 ** |
| Bounce Rate | −0.541 | | | 0.000 ** |
| Average Pages/Visit | −0.763 | | | 0.000 ** |
| Average Time on Site | 1.776 | 1.00 | - | 0.000 ** |
| Unique Visitors | 2.504 | | | 0.000 ** |
| Returning Visitors | −3.160 | | | 0.000 ** |

** Indicate statistical significance at the 99% level.

**Table 8.** Impact of Mobile Engagement metrics on Cryptocurrency websites' Organic Traffic.

| Variables | Standardized Coefficient | $R^2$ | F | *p* Value |
|---|---|---|---|---|
| Constant | 0 | | | 0.000 ** |
| Bounce Rate | 1.037 | | | 0.000 ** |
| Average Pages/Visit | −1.095 | | | 0.000 ** |
| Average Time on Site | 0.217 | 1.00 | - | 0.000 ** |
| Unique Visitors | 2.555 | | | 0.000 ** |
| Returning Visitors | −1.665 | | | 0.000 ** |

** Indicate statistical significance at the 99% level.

Next, in Tables 9 and 10, we show the regressions of cryptocurrency websites' global rank and organic traffic with 4 web traffic sources (direct, paid, social, and search traffic). This time, both regressions don't appear to have significant levels of F-statistic, with *p*-levels higher than 0.05 and 0.01. Apart from the insignificant statistical results, none of the independent variables of traffic sources causes any significant variation to cryptocurrency websites' global rank of organic traffic, leading to the rejection of Hypotheses 3 and 4, which assume that traffic sources significantly affect global rank and organic traffic of cryptocurrency websites. Although the combination of web traffic sources cannot provide sufficient data to define the variation of either global rank and/or organic traffic of cryptocurrency websites, the individual effect of some traffic sources provides a better explanation of the dependent variables. For instance, as we see in Table 11, for the global rank variable, none of the referred traffic sources correlates at a significant rate. Still, when it comes to organic traffic, most traffic sources, except paid traffic, correlate at significant levels with organic traffic. This verifies research Hypothesis 5 regarding organic traffic, as it gets significantly affected by most web traffic sources (direct, referral, social, and search traffic), whereas the global rank variable's variation is not adequately and significantly explained by any web traffic source.

**Table 9.** Impact of Website Traffic Source on Cryptocurrency websites' Global Rank.

| Variables | Standardized Coefficient | $R^2$ | F | *p* Value |
|---|---|---|---|---|
| Constant | 0 | | | 0.75 |
| Direct Traffic | 6.517 | | | 0.614 |
| Paid Traffic | −0.238 | | | 0.901 |
| Social Traffic | −6.863 | 0.689 | 0.554 | 0.504 |
| Search Traffic | 0.396 | | | 0.934 |
| Constant | 0 | | | 0.75 |

**Table 10.** Impact of Website Traffic Source on Cryptocurrency websites' Organic Traffic.

| Variables | Standardized Coefficient | $R^2$ | F | *p* Value |
|---|---|---|---|---|
| Constant | 0 | | | 0.558 |
| Direct Traffic | 0.118 | | | 0.989 |
| Paid Traffic | 0.179 | 0.846 | 1.379 | 0.894 |
| Social Traffic | −0.443 | | | 0.942 |
| Search Traffic | 1.153 | | | 0.74 |

**Table 11.** Pearson correlations of Traffic Sources with Cryptocurrency websites' Global Rank and Organic Traffic.

| Variables | Pearson Stat. | *p* Value |
|---|---|---|
| **Global Rank** | | |
| Constant | −0.010 | 0.985 |
| Direct Traffic | 0.007 | 0.989 |
| Paid Traffic | 0.488 | 0.326 |
| Social Traffic | −0.131 | 0.805 |
| Search Traffic | −0.068 | 0.899 |
| **Organic Traffic** | | |
| Direct Traffic | 0.900 | 0.015 * |
| Referral Traffic | 0.898 | 0.015 * |
| Paid Traffic | 0.499 | 0.314 |
| Social Traffic | 0.880 | 0.021 * |
| Search Traffic | 0.902 | 0.014 * |

* Indicate statistical significance at the 95% level.

The outcomes of the regression analysis indicate some specific patterns. Those patterns have to do with the device visitors use to enter cryptocurrency websites, as well as the engagement metrics of each device. The results show that mobile and desktop devices have a significant effect (significant *p*-levels) on cryptocurrency websites' global rank and organic traffic. The same applies to the engagement metrics (bounce rate, average pages/visit, average time on site, unique and returning visitors). In particular, cryptocurrency websites' organic traffic rises as direct referral, social, and search traffic increases, as well as average time on site and unique visitors of both devices increase. In contrast, average pages per visit and returning visitors decrease, with the bounce rate of mobiles decreasing and desktops increasing. As for global rank, where values near 1 are considered an improvement, it does not get affected by any traffic source visitors may source to reach cryptocurrency websites. Concerning desktop devices, the higher the bounce rate, the more pages visitors see, the fewer time visitors spend on-site, the fewer unique visitors, and the more returning visitors cryptocurrency websites have. Mobile devices provide a better (lower) global rank for cryptocurrency websites. The more bounce rate and pages per visit increase, the fewer time visitors spend on-site, the more the unique visitors, and the fewer the returning visitors are.

### 3.1. Development of Diagnostic and Exploratory Model

After spotting and collecting the required data from the research sample, our next goal is to harvest them in favor of developing an adaptable exploratory model. Positive and negative correlations of web metrics are capitalized in this stage to deploy an efficient and credible model. Abiding by the necessary conditions of data linearity and variables' normality, the authors perform analogous statistical tests for beginning the diagnostic model development. Next, the authors break down the most significant correlations of web metrics by examining the stronger ones between the chosen metrics, which refer to cryptocurrency websites' global rank, organic, direct, referral, search, social and paid traffic, returning and unique customers, bounce rate, average time on site, and pages per customer.

Furthermore, additional knowledge can be extracted from deploying estimation and prediction models. In this stage, the authors developed an exploratory model by utilizing the Fuzzy Cognitive Map Models to show a macro-level analysis of the paper's research environment, including all referred metrics. Through Fuzzy Cognitive Mapping, the entity of the relationships inside a specific environment can be depicted [64], with units/variables intercorrelations being illustrated by an ordering system, providing information about their interactions. FCM depicts variables' negative and positive relationships with blue and red arrows, referring to their cause-and-effect relationships, with thicker arrow lines meaning stronger correlations.

With the diagnostic model developed, as shown in Figure 2, the authors proceed to deploy a micro-level model for a more accurate depiction of the paper's research context. This enables us to locate and predict the most significant relations and coefficients between the research's variables (web metrics). FCM uses quantitative weights for showing component relationships unambiguously and concisely, thus, deploying an organization's macro standpoint regarding digital promotion strategies.

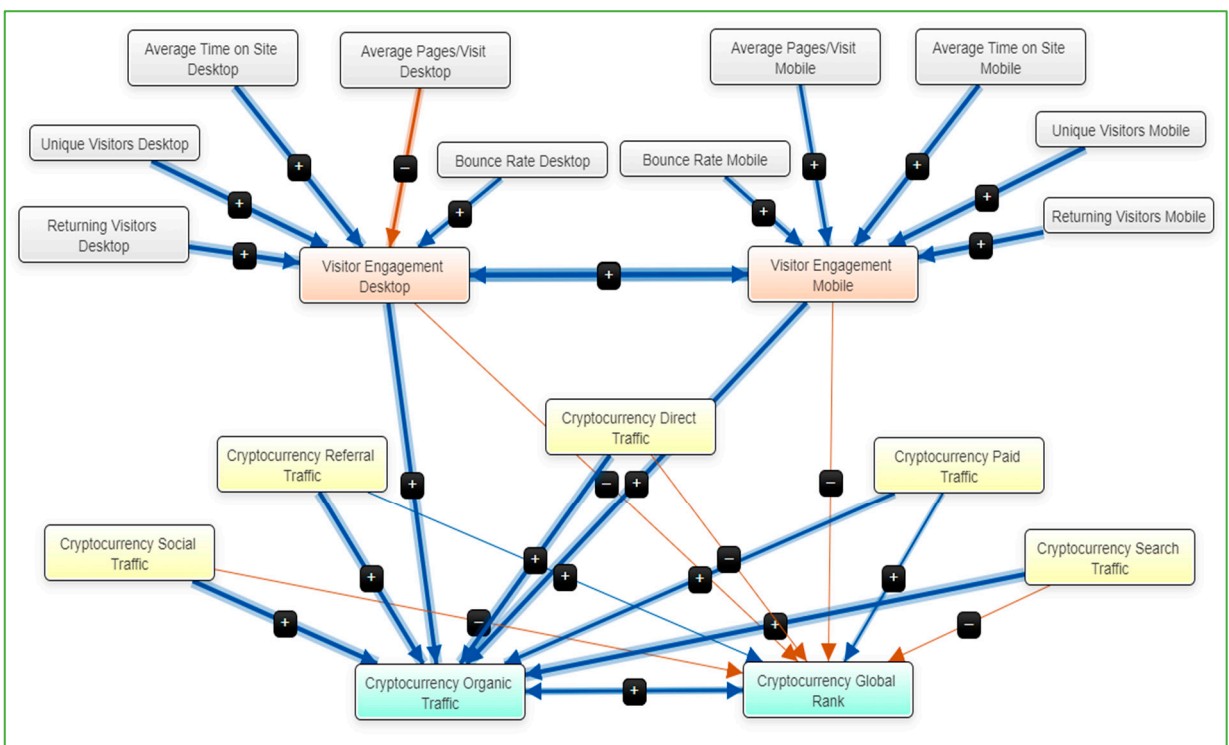

**Figure 2.** Fuzzy Cognitive Mapping depiction of macro-level approach.

Agent-Based Models (ABM), which follow FCM modeling in Section 3.1, provide compelling predictive results, with the agents' dynamic connections being presented accurately [65]. Usage of ABM in the paper's research will allow us to estimate cryptocurrency trade website customers' engagement impact on their web visibility, explained by the global rank and organic traffic metric, in a 360-days simulation period.

### 3.2. Development of Predictive and Simulation Model

The main analysis of the current session is the micro-level one, which is embodied by the Agent-Based Model development (ABM). Through metrics' dynamic correlations, various volatile outcomes are produced, which ABM analysis aims to utilize in a more consistent predictive model [65]. Authors mainly focus on the impact of device preference and their engagement metrics, as well as web traffic sources' implications on cryptocurrency websites' global rank and organic traffic. ABM modeling appears as predefined teams of

agents (people) that interact, via a specific system, producing multiple benefits for decision-making procedures [69] by providing a handful of intel to the micro-level environment regarding its consisting variables. Model's agents follow exactly predefined instructions, set by specific parameters, user-operated, with their interaction based on common operators like and, if, etc. In this stage, regression coefficients and correlations are used to build the model.

Cryptocurrency organizations can be benefited from applying agent-based analysis to their website's engagement and traffic metrics for various important decisions. Organizations performing micro-level analyses of their environment should focus on aligning the desired metrics and variables with the appropriate KPIs. In this way, cryptocurrency organizations can set records for specific web metrics' observations and assess their variation over time. So, applying Agent-Based Modelling can enhance customer engagement and device preference strategy, with advertisement and traffic costs kept low.

The input of regression results regarding web metric interplay, which refers to coefficients and correlations, boosts the ABM model's versatility and effectiveness. Through research [65], our agent-based model is deployed in 360-observation days, using one-time snapshot measurement, as seen in Figure 3. Each agent's movement is unique and is defined by specific parameters' variation. The Java rout used for performing the model below can be seen at Table A1 in the Appendix A section.

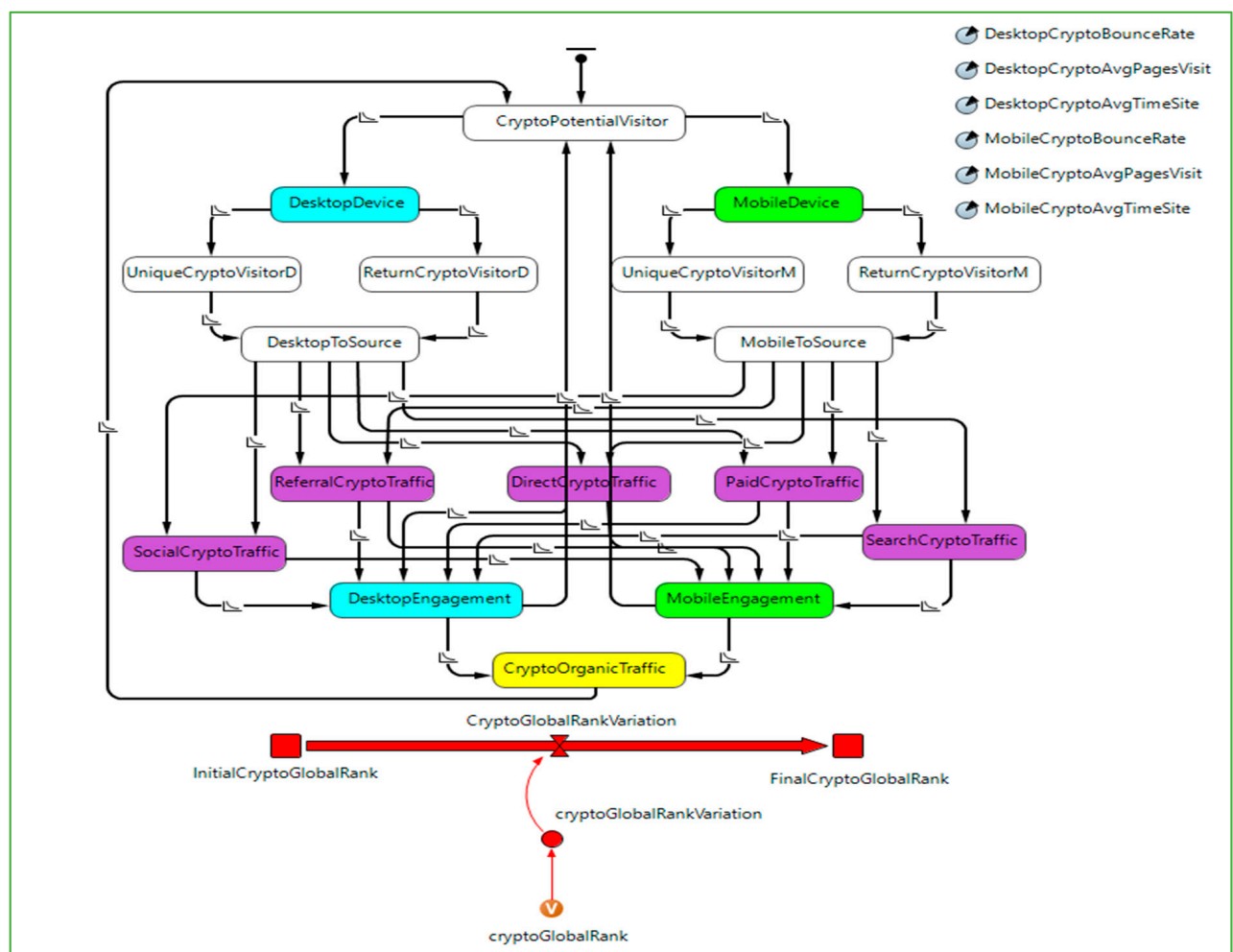

**Figure 3.** Agent-based model deployment for predicting cryptocurrency visitor engagement influence on global rank and organic traffic.

Figure 3 denotes the start of the AB model's process from the cryptocurrency potential visitors statechart. Potential website visitors land on a cryptocurrency site via desktop or mobile devices, and due to entrance repetition they belong either to unique or returning visitors statecharts. Then, having calculated the means of website entrance, the model seeks to calculate the web traffic source that visitors choose to enter cryptocurrency websites. These sources refer to referral, direct, paid, social, and search traffic statecharts, and since visitors enter the cryptocurrency website, their engagement is calculated in a desktop or mobile statechart using the parameters shown in the upper right portion of Figure 3. Both engagement statecharts end organic cryptocurrency traffic statecharts and cause variation in their global rank through a dynamic system. The model's simulation operation produces the allocation of agents seen in Figure 4, where the fluctuation of the paper's web metrics is presented.

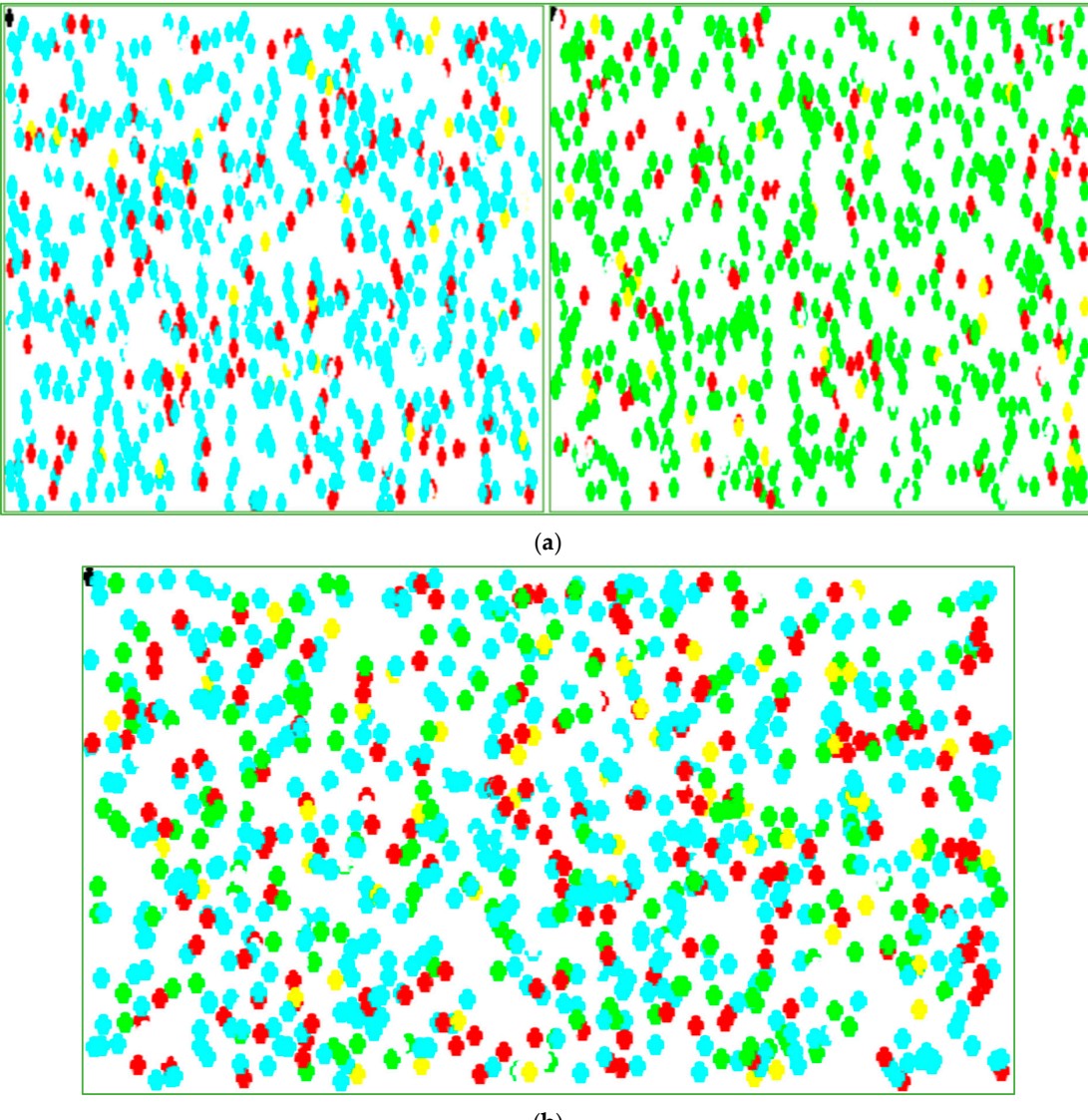

(**a**)

(**b**)

**Figure 4.** Population allocation in the experiment with 1000 agents. (**a**) individual devices' engagement levels and organic traffic. (**b**) combined devices' engagement levels and organic traffic.

In the model's period of 360-days, the dispersion of 1000 agents looks like Figure 4. This figure consists of agents that use desktop and mobile devices and get engaged with the cryptocurrency websites they visit, thus increasing its organic traffic. White represents agents that potentially would enter a cryptocurrency website, cyan represents agents using

desktop devices, green those who use mobile devices, red the engagement percentage of those devices, and yellow the number of agents that constitute cryptocurrency organic traffic. (a) In the first stage of Figure 4, we get the allocation of agents that individually use desktop or mobile devices and their engagement level and organic traffic percentage of cryptocurrency websites. (b) Having analyzed both devices' usage, we get slightly increased engagement levels and organic traffic percentage. We can see that the combined device approach in terms of digital promotion can cause increased customer engagement and organic traffic to cryptocurrency websites.

In Figure 5, we get the results of the paper's variables of interest over 360-simulation days. The period can be seen on the horizontal axis of the figure, while on the vertical one, the numerical results of desktop and mobile engagement, global rank, and organic traffic are shown. The simulation process has produced the following results. Rising levels of desktop and mobile device engagement tend to decrease cryptocurrency websites' global rank and increase their organic traffic, thus contributing to enhanced website visibility and digital brand name. Generally, cryptocurrency organizations should focus on their website customers' metrics since they provide significant insights into their web performance.

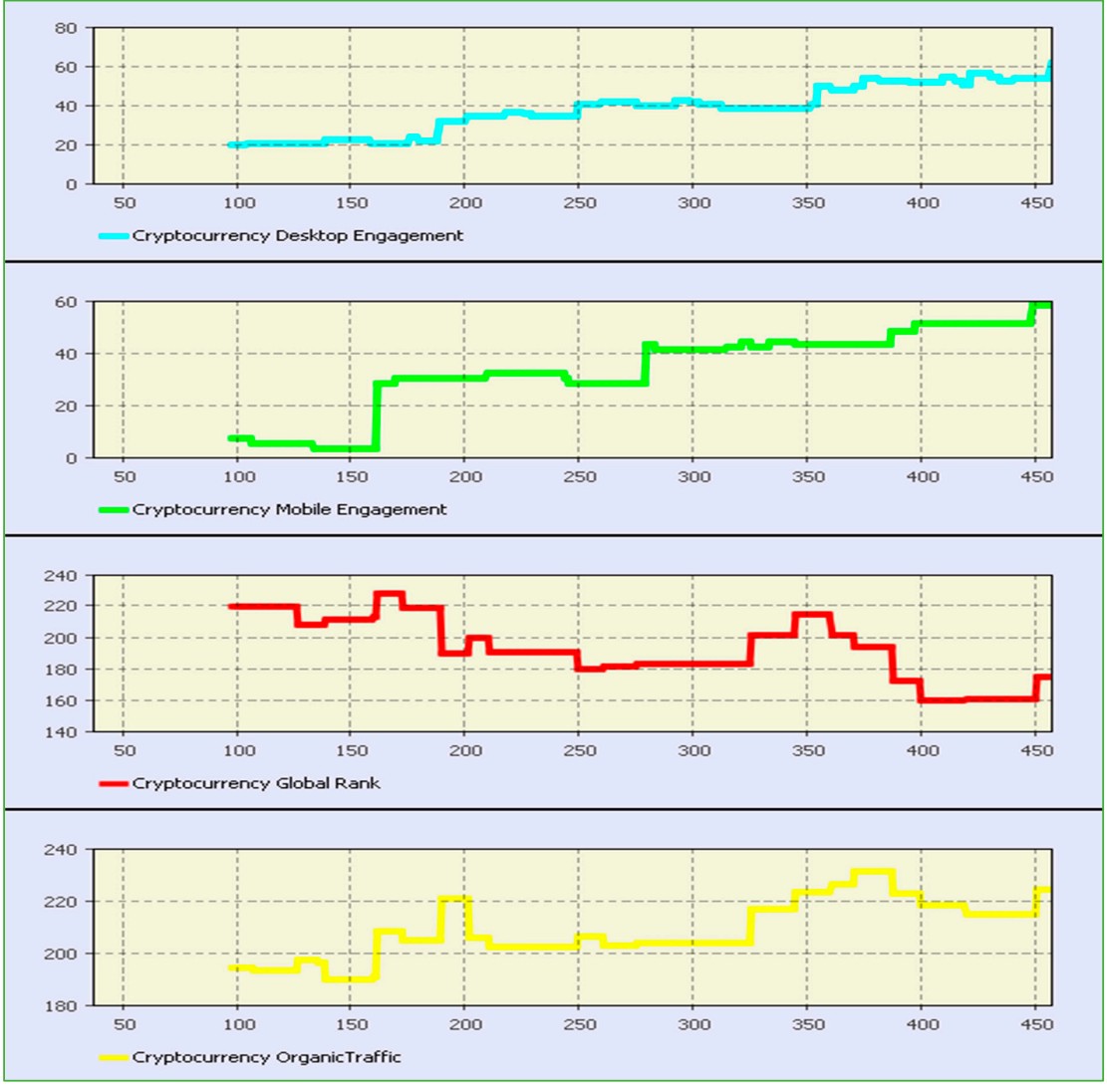

**Figure 5.** Depiction of cryptocurrency websites' desktop/mobile customer engagement and its impact on their global rank and organic traffic.

## 4. Discussion

Increased focus has been granted to elaborating a solid and pioneering methodological context, emphasizing the offering of suitable intelligence concerning cryptocurrency websites' digital promotion. For cryptocurrency websites to gain more web visibility and improve SEO strategies, they should harvest, in their favor, website consumers' behavior through web analytic metrics. For research purposes, the website samples were collected from the 10 most known and highly rising cryptocurrencies in the first half of 2021. To sharpen their digital promotion, cryptocurrency organizations could monitor global rank and organic traffic variations through website customer device preference, engagement, and traffic source selection. In this way, by following this pioneering analysis, cryptocurrency organizations will be able to increase website visibility and digital brand name by attracting more customers.

The factors chosen to represent the benefits for cryptocurrency websites' visibility and digital brand name are global rank and organic traffic. Through regression analysis, global rank varied up to −435.2% and −357.9% (decrease equals enhancement) from rises in mobile and desktop usage, respectively, while neither combined nor individual traffic source selection caused significant variation. Organic traffic varied up to 250.4% and 255.5% regarding desktop and mobile engagement increases, with only individual traffic source selection causing a significant variation since combined sources' impact was marked as negligible. Hence, cryptocurrency organizations need to increase website customers' engagement to gain more organic traffic and enhance global rank, leading to higher web visibility and digital brand name [70].

## 5. Conclusions

### 5.1. Conclusions on Cryptocurrency Websites Digital Promotion Based on Device Preference, Customer Engagement, and Traffic Sources

The main conclusions of this research are the efficiency of cryptocurrency organizations' digital promotion reliance on their website customers' behavior. This provides organizations with insights that suggest website customers' engagement metrics and device preferences. Cryptocurrencies are based on innovative blockchain technology, and organizations that issue and support them could benefit from efficient promotion. Blockchain technology, in the sphere of international marketing and advertising, offers a much more comprehensive perspective for marketing parameters, such as the duration and effectiveness of commercials, and enhances visibility for promoting organizations [52].

In this paper, the impact of website customers' engagement metrics on cryptocurrency organizations' global rank and organic traffic was shown to be significant. The research has also shown that both mobile and desktop devices can affect the outcome of cryptocurrency organizations' digital promotion strategies, examined through website customers' behavior and web analytic metrics. In the same way, it has been discovered that combined website traffic source promotion campaigns do not significantly affect cryptocurrency organizations' digital brand names. Thus, an individual web traffic source strategy focused on a one-time source (e.g., direct traffic) can provide enhanced results to cryptocurrency organizations' digital promotion by significantly affecting their organic traffic.

Based on similar studies [71], emphasis has been given to the role of harvesting web analytics in favor of digital promotion strategies focused on cryptocurrency organizations. There is already huge potential for blockchain applications to be adopted by several market sectors, such as finance, supply chain management, digital promotion, and marketing. The need for organizational strategy performance measurement of such applications is rising [72]. Engaging more customers is one of the most important targets of digital promotion and advertisements and may lead to an organization's increased digital brand name, leading to profitability. Assessing digital promotion and advertisement campaigns distinguish the importance of web analytic metrics [73] and customers' engagement levels [74]. The authors deployed a three-stage methodology consisting of setting the required KPIs, gathering the data, and running the necessary regression analysis.

*5.2. Research Implications*

There are several practical implications originating from the paper's results. Which digital promotion strategies can produce a better result for cryptocurrency organizations' digital brand names have been widely examined. Determinants of cryptocurrency organizations' digital brand names were selected from the organic traffic and global rank metrics. For the various strategies tested, web analytic metrics of mobile and desktop customer engagement and traffic sources were utilized. Results of the regression and correlation analysis extrapolate that performed regressions significance indicate the magnitude of the dependent variables' effect caused by the independent ones, as $R^2$ statistics adorn around 1.00 [75].

Digital branding of cryptocurrency organizations could potentially be elaborated by web analytics [70,76], based on appropriate KPIs selection. More specifically, the distinction between mobile and desktop web analytic metrics can provoke different magnitude effects on organizations' digital brand names [18]. The same can be said about the type of website traffic source customers choose to enter a site [17]. Instead of a combined web traffic sources strategy, our research has shown that cryptocurrency organizations should prefer to focus more on one source at a time to advertise and attract customers. Hence, the regular observation of mobile and desktop customers' engagement metrics and website traffic sources, except paid traffic, should be carried out.

Given the importance of web analytics in explaining consumers' website behavior, implication and examination of cryptocurrency organizations' website analytics could be proven advantageous. Besides the application nature of the results, this analysis shows that website engagement levels of customers highly affect the digital marketing performance of cryptocurrency websites. This research shows that the more websites succeed in creating interesting layouts and simplifying the search process for visitors using both desktop and mobile devices, the more visitor engagement and loyalty increase as the digital marketing campaigns' efficiency attracts more organic visitors. By increasing the organic website's visits, an organization can reduce the expenses of digital marketing campaigns and increase its sales, leading to enhanced profitability.

As an immense effect, it is suggested that cryptocurrency organizations that aim to sharpen their digital promotion campaigns should consider monitoring key website metrics. Such metrics can efficiently demonstrate the variation of organic traffic and global rank, important indicators of the corporate digital brand name. The applied methodology supplies contingent cryptocurrency organizations' promotion strategists with requisite functional information concerning the fruitfulness of particular digital promotion strategies. Research's outcomes distinguish customers' engagement metrics both on mobile and desktop devices as a determinant factor of cryptocurrency organizations' brand name enhancement and an area for further improvement.

*5.3. Limitations*

With respect to this research, a few insights regarding its limitations can be provided. For the development of the model and the analysis of the outcomes, specific metrics of website users' behavior were utilized and collected for 180 days. To obtain a deeper understanding of the website behavioral analytics on cryptocurrency organizations' digital marketing, more website analytic metrics should be examined, such as the technical factors of these websites.

*5.4. Future Work*

Throughout this paper, the research interest centers around the efficiency of cryptocurrency organizations' digital promotion strategies. For this reason, specific KPIs of website analytic metrics and customer behavior were chosen. The deployed methodology could be adopted by the supply chain management and green logistics sectors [70], apart from the cryptocurrency market, as well as by the financial and banking sectors. Such adoption could

be enhanced by implementing Neuromarketing tools, which provide valuable information regarding customers' brain activity [77], capable of explaining multiple behavioral patterns.

**Author Contributions:** Conceptualisation, D.P.S. and N.T.G.; methodology, N.T.G.; software, N.T.G.; validation, D.P.S., N.K. and C.T.; formal analysis, D.P.S. and C.T.; investigation, N.T.G. and N.K.; resources, N.T.G.; data curation, N.T.G. and N.K.; writing—original draft preparation, N.T.G.; writing—review and editing, N.T.G. and N.K.; visualization, D.P.S. and N.T.G.; supervision, D.P.S. and C.T.; project administration, D.P.S. and N.T.G.; funding acquisition, D.P.S. and N.K. All authors have read and agreed to the published version of the manuscript.

**Funding:** This research received no external funding.

**Institutional Review Board Statement:** Not applicable.

**Informed Consent Statement:** Not applicable.

**Data Availability Statement:** Not applicable.

**Conflicts of Interest:** The authors declare no conflict of interest.

## Appendix A

**Table A1.** JAVA coding route for defining cryptocurrency organizations' organic traffic and global rank.

| JAVA coding route for defining cryptocurrency organizations' organic traffic and global rank |
|---|
| case CryptoPotentialVisitor://(Simple state (not composite))<br>statechart.setActiveState_xjal(CryptoPotentialVisitor);<br>{<br>potentialCryptoVisitor++L<br>;}<br>transition.start();<br>transition1.start();<br>return;<br>case DesktopDevice://(Simple state (not composite))<br>statechart.setActiveState_xjal(DesktopDevice);<br>transition2.start();<br>transition3.start();<br>return;<br>case UniqueCryptoVisitorD://(Simple state (not composite))<br>statechart.setActiveState_xjal(UniqueCryptoVisitorD);<br>{<br>uniqueCryptoDesktopVisitor = potentialCryptoVisitor<br>;}<br>transition6.start();<br>return;<br>case DesktopToSource://(Simple state (not composite))<br>statechart.setActiveState_xjal(DesktopToSource);<br>transition10.start();<br>transition11.start();<br>transition12.start();<br>transition13.start();<br>transition14.start();<br>return;<br>case SocialCryptoTraffic://(Simple state (not composite))<br>statechart.setActiveState_xjal(SocialCryptoTraffic);<br>{<br>cryptoSocialTraffic++<br>;}<br>transition15.start();<br>transition27.start(); |

**Table A1.** *Cont.*

| JAVA coding route for defining cryptocurrency organizations' organic traffic and global rank |
| --- |

```
return;
case DesktopEngagement://(Simple state (not composite))
statechart.setActiveState_xjal(DesktopEngagement);
{
desktopEngagement = (1-DesktopCryptoBounceRate) + DesktopCryptoAvgPagesVisit +
DesktopCryptoAvgTimeSite + uniqueCryptoDesktopVisitor + returnCryptoDesktopVisitor
;}
transition25.start();
transition32.start();
return;
case CryptoOrganicTraffic://(Simple state (not composite))
statechart.setActiveState_xjal(CryptoOrganicTraffic);
{
cryptoGlobalRank = cryptoGlobalRankMobile + cryptoGlobalRankDesktop;
cryptoOrganicTraffic = cryptoOrganicTrafficDesktop + cryptoOrganicTrafficMobile
;}
transition34.start();
return;
case MobileEngagement://(Simple state (not composite))
statechart.setActiveState_xjal(MobileEngagement);
{
mobileEngagement = (1-MobileCryptoBounceRate) + MobileCryptoAvgPagesVisit +
MobileCryptoAvgTimeSite + returnCryptoMobileVisitor + uniqueCryptoMobileVisitor
;}
transition26.start();
transition33.start();
return;
case ReferralCryptoTraffic://(Simple state (not composite))
statechart.setActiveState_xjal(ReferralCryptoTraffic);
{
cryptoReferralTraffic++
;}
transition16.start();
transition31.start();
return;
case SearchCryptoTraffic://(Simple state (not composite))
statechart.setActiveState_xjal(SearchCryptoTraffic);
{
cryptoSearchTraffic++
;}
transition18.start();
transition28.start();
return;
case PaidCryptoTraffic://(Simple state (not composite))
statechart.setActiveState_xjal(PaidCryptoTraffic);
{
cryptoPaidTraffic++
;}
transition19.start();
transition29.start();
return;
case DirectCryptoTraffic://(Simple state (not composite))
statechart.setActiveState_xjal(DirectCryptoTraffic);
{
cryptoDirectTraffic++
;}
transition17.start();
transition30.start();
```

**Table A1.** *Cont.*

| JAVA coding route for defining cryptocurrency organizations' organic traffic and global rank |
|---|
| ```
return;
case ReturnCryptoVisitorD://(Simple state (not composite))
statechart.setActiveState_xjal(ReturnCryptoVisitorD);
{
returnCryptoDesktopVisitor = potentialCryptoVisitor
;}
transition7.start();
return;
case MobileDevice://(Simple state (not composite))
statechart.setActiveState_xjal(MobileDevice);
transition4.start();
transition5.start();
return;
case UniqueCryptoVisitorM://(Simple state (not composite))
statechart.setActiveState_xjal(UniqueCryptoVisitorM);
{
uniqueCryptoMobileVisitor = potentialCryptoVisitor
;}
transition8.start();
return;
case MobileToSource://(Simple state (not composite))
statechart.setActiveState_xjal(MobileToSource);
transition20.start();
transition21.start();
transition22.start();
transition23.start();
transition24.start();
return;
case ReturnCryptoVisitorM://(Simple state (not composite))
statechart.setActiveState_xjal(ReturnCryptoVisitorM);
{
returnCryptoMobileVisitor = potentialCryptoVisitor
;}
transition9.start();
return;
default:
super.enterState(_state, _destination);
return;
}
}
``` |

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
