# Peer review of "Digital Marketing Enhancement of Cryptocurrency Websites through Customer Innovative Data Process"

_processes, doi:10.3390/pr10050960_

Round 1
Reviewer 1 Report
The authors of "Digital Marketing Enhancement of Cryptocurrency Websites through Innovative Customer Data Process" present a relevant topic, namely "methods of promoting websites with cryptocurrencies" through Big Data and web analysis by research authors, which it is found globally on the decentralization of the payment system with a direct impact on the banking financial system, which makes the work have the effect of multiplying the level of financial governance at the level of institutions that use such financial instruments.
The bibliographic sources, citations, key concepts used in the study are appropriately mentioned and presented by the authors of the research. Specifically, those that show that "the value of Bitcoin is influenced by a wide range of factors, including global opinion, media and buzz [2]", as well as the fact that "cryptocurrencies are a relatively new payment option that offers a competitive advantage for companies' websites [15]”.
The research methodology is presented based on variables, web Analytics / KPIs and global ranking (web) / month. The authors of the research base their research results starting from 5 working hypotheses, mainly oriented towards the impact of cryptocurrencies. Web analytics tools, and the correlations and relationships between sample variables (a sample of 3,000,000 behavioral website data) are tools used by authors to analyze data in the paper. The model used by the authors is that of linear regression modeling.
The results of the research are presented by the authors of the research based on the models mentioned in the research methodology, highlighting "the vices for mobile devices,… on the global ranking of websites with cryptocurrencies…. and for the values ​​of involvement (rejection rate, average pages / visit, average time on site, unique and returning visitors) ”. Furthermore, the authors argue through web analytics metrics the impact on website consumer behavior. However, we ask the authors of the research to highlight the main scientific results as a personal contribution to the scientific literature, given that the results are adequately presented but much focused on the application side and can create a multiplier effect on financial governance in the digital space of cryptocurrencies.
The conclusions are presented by the authors of the research, respectively the authors emphasize that "the impact of the values ​​of customer involvement of the website on Crypto and on the global ranking of monetary organizations and on organic traffic was significant". Furthermore, we suggest that the study authors also present the limitations of the study. Future research is adequately presented by the research team. However, as we mentioned in the results chapter, we appreciate that the personal scientific results that contribute to the scientific literature should be highlighted.
We congratulate the research team, we suggest the revision of the paper according to the above mentioned, and after the revision we propose for acceptance the paper.
Author Response
On behalf of the authors, I would like to express our gratitude for your suggestions and for working on optimizing our paper.

Reviewer 2 Report
In this paper, the authors propose a three-stage models for analyzing the role of cryptocurrencies and their impact on customers. The idea they propose is interesting but there are some aspects that need to be improved in the paper before its publication.
The most important aspect that needs to be improved concerns the overall objective of the paper in the context of the existing literature. To make this understood, the authors should restructure Section 1 differently than it is now. In particular, they should devote an initial subsection to motivations: Why is an approach like theirs useful? What are the general problems that it wants to solve?
Next, they should show a subsection devoted to related literature: What approaches in this context have been proposed? Why is an approach like theirs needed in this scenario? What does it have over existing approaches?
In a third subsection, the authors should describe their approach. Before making an analytical description with a set of question that the approach intends to answer (as it currently is), the authors should provide a general description (e.g., through a block diagram) of the proposed approach. In the current version of the paper, such a description can be found in part in Figure 1, but should be represented in a more abstract way by untying it from the assumptions proposed by the authors.
A second important aspect to improve concerns the related literature. Indeed, the authors should consider some recently published papers in the field of blockchains. For example, they should mention at least the following papers (very related to their own), along with other recent ones: "A Social Network Analysis based approach to investigate user behavior during a cryptocurrency speculative bubble", "Defining user spectra to classify Ethereum users based on their behavior".
I really appreciated the experimental part of the paper, which I consider rich and comprehensive.
Last but not least, the authors should correct various typos in the paper. For example, "Oday" I think should be "Today".
Author Response

(The authors gave the same response as above.)

Round 2
Reviewer 2 Report
The authors have addressed my previous concerns. I think that the paper can be accepted.